# Evolution of Telencephalon Anterior–Posterior Patterning through Core Endogenous Network Bifurcation

**DOI:** 10.3390/e26080631

**Published:** 2024-07-26

**Authors:** Chen Sun, Mengchao Yao, Ruiqi Xiong, Yang Su, Binglin Zhu, Yong-Cong Chen, Ping Ao

**Affiliations:** 1Center for Quantitative Life Sciences & Physics Department, Shanghai University, Shanghai 200444, China; sunchen@shu.edu.cn (C.S.); eric_yao@shu.edu.cn (M.Y.); ruiqixiong@shu.edu.cn (R.X.); suyang@shu.edu.cn (Y.S.); zhubinglin@shu.edu.cn (B.Z.); 2School of Biomedical Engineering, Sichuan University, Chengdu 610065, China

**Keywords:** telencephalon, evolution, free energy principle, gene regulatory network, endogenous network theory, nonlinear process

## Abstract

How did the complex structure of the telencephalon evolve? Existing explanations are based on phenomena and lack a first-principles account. The Darwinian dynamics and endogenous network theory—established decades ago—provides a mathematical and theoretical framework and a general constitutive structure for theory–experiment coupling for answering this question from a first-principles perspective. By revisiting a gene network that explains the anterior–posterior patterning of the vertebrate telencephalon, we found that upon increasing the cooperative effect within this network, fixed points gradually evolve, accompanied by the occurrence of two bifurcations. The dynamic behavior of this network is informed by the knowledge obtained from experiments on telencephalic evolution. Our work provides a quantitative explanation for how telencephalon anterior–posterior patterning evolved from the pre-vertebrate chordate to the vertebrate and provides a series of verifiable predictions from a first-principles perspective.

## 1. Introduction

As the material basis of cognitive abilities, the telencephalon results from hundreds of millions of years of evolution [1,2,3]. A series of events, like genome duplication and the emergence of new genes [4,5], the segregation of brain functions and the establishment of effective connectivity [6,7,8], increases in core gene network interconnectivity [9], and neuropathway duplication [2], provide phenomenological explanations for how the complex structure of the telencephalon has evolved to this day. However, just like the research on celestial motion before the introduction of Newtonian mechanics, these phenomenon-based explanations lack a fundamental component: a first-principles account.

The free energy principle guides the brain to achieve cognitive optimization through minimizing prediction errors, providing an explanation for specific instances of brain function [6,7,10,11]. Recently, some researchers have expressed concerns that research in the life sciences pays more attention to the experiment, rather than theory–experiment coupling [12,13,14], or even features a circling back aspect, at least in cancer research, after half a century [15]. Indeed, the phenomena of life are full of complexity, and complexity science itself is a newborn area. The discovery of a first principle in life is bound to be difficult. Nonetheless, there are fruits borne in this “circle”. Decades after the introduction of Boolean dynamics into the gene network [16,17], the analysis of the dynamical structure of the hydrogen model in life science, the phage-λ gene switch [18], provided an example of the application of a general law, along with a mathematical framework for the evolution dynamics [19,20]. Following this approach, the ensuing theory provides a general constitutive structure of evolution dynamics at the gene network level. The endogenous network theory [21] was established years after the introduction to system biology [22] of the landscape concept raised by Wright and Waddington, allowing qualitative and quantitative comparisons between theories and experiments [23,24]. Under these theoretical constructs, a series of efforts were made to address questions about cancer and development [25]. Now, we may have an apt mathematical and theoretical framework—as well as the knowledge obtained from experiments—to address the question of how our brains evolved at the gene network dynamics level in a first-principles way.

In this work, an attempt is made, by revisiting a pre-constructed model [26,27,28] of telencephalic anterior–posterior patterning in vertebrates containing 5 nodes and 10 edges, to demonstrate that the dynamic behavior of such a relatively small network can answer a key question in evolutionary neurodevelopment, under the mathematical and theoretical framework described above.

Specifically, based on the above theories and frameworks, we establish a methodology to reconstruct plausible evolutionary trajectories, with a focus on the interactions between the expression of genes relevant to the anterior–posterior patterning of the telencephalon. We used a well-validated theoretical model constructed by Giacomantonio and Goodhill [26,27,28], which exhibits relatively clear dynamic behavior. From a dynamical perspective, we expect a system capable of generating anterior–posterior telencephalon patterning to have at least two stable attractors, representing the gene-expression profiles at each of the poles of this axis. In this work, we use Boolean dynamics and modified equation forms to explore different variants of gene-expression interactions. We find that there are always two attractors in all variants, which are robust to the equation form, indicating that small variations in evolution can lead to similar phenomena, with the formation of anterior–posterior patterning in the telencephalon. Furthermore, through an analysis of bifurcation patterns in a continuous dynamical system approximating the Boolean dynamics, the evolutionary trajectory is extrapolated back to the telencephalon formed by the absence of anterior–posterior patterning. Specifically, we assume that evolution changed the parameters of the gene-expression interaction dynamics and that, at some stage in evolution, there must have been a bifurcation in these dynamics from one to two stable attractors to explain the transition to a telencephalon with anterior–posterior patterning.

## 2. Materials and Methods

### 2.1. Boolean Dynamics

For the unaltered function set, we used “AND” to represent activating interaction, and “NOT” to represent inhibitory interaction. These interactions define causal dynamics among gene expression. Some examples are given below:
A = Fgf8 A (t + 1) = D (t)B = Emx2 B (t + 1) = NOT A (t)C = Pax6 C (t + 1) = A (t) AND (NOT B (t)) AND (NOT E (t))D = Sp8 D (t + 1) = (NOT B (t)) AND (NOT E (t))E = Coup-tfi E (t + 1) = (NOT A (t)) AND (NOT C(t)) AND (NOT D (t))

For genomic variants, a variable affected by 1, 2, and 3 variables will result in 1, 2, and 8 equations, respectively. Thus, we generated 1 × 1 × 8 × 2 × 8 = 128 sets of functions. All variants are presented in the Appendix A.

We iterated the function set with initial vectors until stable, then collected and counted attractors, attractor types, and basin of attraction sizes. The detailed results are presented in the Appendix A.

### 2.2. ODE

The ODE function we used is as follows:$$x(1)=\mathrm{Fgf}8;\;\frac{dx\left(1\right)}{dt}=-x\left(1\right)+\frac{k*x{\left(5\right)}^{h}}{1+k*x{\left(5\right)}^{h}}$$
$$x(2)=\mathrm{Pax}6;\;\frac{dx\left(2\right)}{dt}=-x\left(2\right)+\frac{k*x{\left(1\right)}^{h}}{1+k*x{\left(1\right)}^{h}}*\frac{1}{1+k*\left(x{\left(3\right)}^{h}+x{\left(4\right)}^{h}\right)}$$
$$x(3)=\mathrm{Emx}2;\;\frac{dx\left(3\right)}{dt}=-x\left(3\right)+\frac{1}{1+k*x{\left(1\right)}^{h}}$$
$$x(4)=\mathrm{Coup}\text{-}\mathrm{tfi};\;\frac{dx\left(4\right)}{dt}=-x\left(4\right)+\frac{1}{1+k*\left(x{\left(1\right)}^{h}+x{\left(2\right)}^{h}+x{\left(5\right)}^{h}\right)}$$
$$x(5)=\mathrm{Sp}8;\;\frac{dx\left(5\right)}{dt}=-x\left(5\right)+\frac{1}{1+k*\left(x{\left(3\right)}^{h}+x{\left(4\right)}^{h}\right)}$$

We used Newton’s iteration to obtain fixed points; determined the stability of the fixed point using the real part of the eigenvalue of its Jacobian matrix; and used Euler’s method to verify the stable fixed points.

For the vanilla parameter set (*h* = 3, *k* = 10), we first used 1000 random initial vectors, followed by 100,000 random initial vectors, and both resulted in the same fixed points, so 1000 random initial vectors were used to explore the parameter space.

The connection between transition states and stable states was obtained by perturbing the transition state with a small random vector; in this work, the number of such small random vectors was 100.

## 3. Results

### 3.1. A Coarse-Grained Core Endogenous Network for Telencephalon Already Exists

A Boolean five-node network was constructed, which reproduced the experimentally observed gradients of the anterior–posterior patterning of early cortical development in mammals [26,28]. From the perspective of data fitting, the existing experimental data were not sufficient to uniquely specify the interactions of a base network. And if such a base network exists, how the latent factors regulate the base network is not clear [26,27,28].

By way of comparison, Giacomantonio and Goodhill constructed a network with robust dynamic behavior, which is one of the most important features of the “backbone” or “core” structure of an endogenous network—shaped by evolution—that we argued in 2008 is conserved among species during evolution [21,25]. We found that an ortholog exists in amphioxus, one of the closest living vertebrate relatives that diverged from the vertebrate lineage around 550 million years ago, in all five genes [5,29,30] in this network. Nevertheless, Benito-Gutiérrez et al. discovered the resemblance of the expression pattern between the mature amphioxus cerebral vesicle and the developing vertebrate telencephalon [31]. These facts indicate that the network Giacomantonio and Goodhill constructed is conserved in the pre-vertebrate chordate and the vertebrate during evolution.

To further confirm the robustness of the model, we generated 128 Boolean function variations based on the network (Figure 1A) described by Goodhill in 2018 [28] by using a combination of replacing “AND” with “OR”. With the default operators, two point attractors which correspond to the expression pattern of the posterior (boole_p1) or anterior (boole_p2) telencephalon separately, and 2 limit cycles (boole_li7 and boole_li18), were generated by traversing all initial vectors (Figure 1B). For all Boolean variations, we obtained 26 attractors with 4 point attractors and 22 linear attractors (Figure 1B). Among them, two attractors, the posterior-like attractor boole_p1 and the anterior-like attractor boole_p2, were obtained in all Boolean variations (Figure 1E). The proportions of boole_p1 and boole_p2 were equivalently the most common with more than 36% (Figure 1C–E). These results further confirmed the robustness of the Boolean dynamic behavior of this network.

### 3.2. Bifurcations Observed with Evolving Hill Coefficient

Complex dynamic behavior, like transition states and bifurcation, cannot be obtained in Boolean dynamics [17,25,32]. Thus, we turn our attention to the ordinary differential equation (ODE) model. In this work, we used the double-normalized Hill equation, which has been described in detail previously [17,25]. The details of the direct interactions among genes in the current network are still unclear, let alone their dynamic parameters. In order to extract insights from such limited empirical constraints, a set of dimensionless mesoscopic parameters, *h* and *k*, were used in this work.

We chose *h* = 3, *k* = 10 for the first trial of the parameter set (the vanilla set) based on our previous modeling experience [25]. Under this set, we obtained three fixed points, with two stable ones (stable state) and an unstable one (transition state) (Figure 2A,B). The expression patterns of the two stable states (Figure 2A,B; vanilla_sta1, vanilla_sta2) are consistent with the point attractors in the Boolean Dynamics (Figure 2A,B; Boole_p1, Boole_p2). The transition state is also consistent with the average of all the limit cycles in the Boolean dynamics with the unaltered functions (Figure 2A,B; Boole_li_allme: the average of Boole_li_7 and Boole_li_18 in Figure 1B).

In a nonlinear system, bifurcations may occur as the parameters change. Oster and Alberch argued that mathematical bifurcations could be an important mechanism behind development and evolution 40 years ago [33,34]. We therefore explored the parameter space in order to find such bifurcations.

Fitting an elephant with a wiggling trunk was not our goal; therefore, the number of free parameters needed to be reduced. The number of arbitrary parameters in this model was set to 20 (2n, n = edge number). Given that the empirical details of these dynamical parameters are not clear, and genes in the core network level should be considered equally important—as we argued before [21,25]—we used the same parameters for all edges, thereby reducing the free parameter number from 20 to 2. Considering that the concentration/function of genes was already normalized to [0, 1]—in order to center the S shape part of the sigmoid function—we let *k* = 2*^h^*, therefore reducing the number of free parameters to 1.

**Figure 2 entropy-26-00631-f002:**
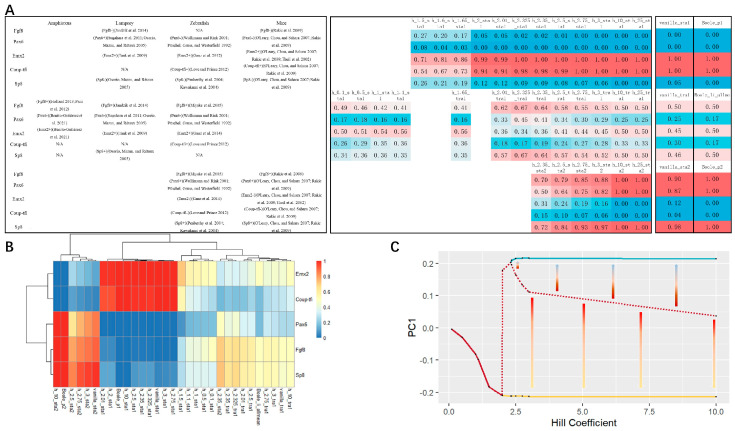
(**A**) Comparison between computation results of different parameters and experimental data. All fixed points under all parameters (*h*) we tried are displayed in the middle table. The four columns on the left are the experimental data of amphioxus [31,35,36], lamprey [37,38,39,40], zebrafish [41,42,43,44,45,46,47,48], and mice [49,50,51]. The two columns on the right show the results obtained under the vanilla parameter set (*h* = 3, *k* = 10) and using unaltered Boolean functions (*h*→ + ∞), and the others are arranged from small to large for *h*. (**B**) Heatmap of all ODE fixed points demonstrated after hierarchical clustering. (**C**) Bifurcations occur with the growth of *h*. Arrows point to the stable states from the transition state (N/A: related experimental data not found; sta: stable state; tra: transition state).

There is an increase in the *h* (*h* = 10, 25; *k* = 2*^h^*) results in two stable states and one transition state, which is similar to the dynamic behavior under the vanilla parameter set (Figure 2A,B). Since the Hill equation is equivalent to its Boolean form, when *h* takes the limit, we expected that the dynamic behavior would not change significantly with further increases in *h*.

We then explored a parameter space with a smaller *h*, and discovered two bifurcations (1.6 < *h* < 1.65; 2.325 < *h* < 2.35; *k* = 2*^h^*). When *h* < 1.6, there is only one stable state, which has pattern similar to the transition state with the vanilla parameter set. The pattern gradually evolves from the vanilla transition state-like pattern to a vanilla stable state1-like pattern with increasing *h*. Passing over the first bifurcation (1.6 < *h* < 1.65), a transition state emerges, which then evolves to a transition state-like pattern similar to vanilla (1.65 < *h* < 2.325). Except for one situation (when *h* = 2), there is only one stable state. The second bifurcation (2.325 < *h* < 2.35) gives birth to the second stable state (Figure 2C).

### 3.3. Evolution of the Telencephalon Anterior Posterior Patterning by Bifurcations of the Network Dynamic Behavior

Remarkably, the dynamic behavior of this network is highly consistent with the facts we know about the evolution of the telencephalon.

A telencephalon-like structure has been identified recently in amphioxus with a low level of Pax4/6 and a high level of EmxA, EmxB [31], and Fgf8 [36] (Figure 2A; amphioxus). Only one stable state exists with weak or negative cooperativity in this network (0 < *h* < 1.6), with a low level of Pax6 and an intermediate–high level of Emx2 and Fgf8, which is consistent with the expression pattern of the telencephalon-like structure identified in amphioxus. The expression pattern of this solitary stable state gradually evolves into a posterior telencephalon-like pattern with increasing *h*.

With a further increase in *h* (1.65 ≤ *h* ≤ 2.325), the first bifurcation occurs, giving birth to a stable state and a transition state. The expression pattern of the stable state is consistent with the posterior part of the telencephalon, and the expression pattern of the ‘popping-out’ transition state is similar to the anterior part of the telencephalon.

This model results in two stable states consistent with telencephalon anterior–posterior patterning in vertebrates after the second bifurcation (*h* ≥ 2.35), and one transition state, which shares a similarity in expression pattern with the telencephalon-like structure in amphioxus as well as the stable state with weak or negative cooperativity in this network. These observations remind us of the biogenetic law [52].

In the validation data of lampreys, zebrafish, and mice, molecular patterns consistent with the theoretical results were identified. For lampreys, there exist two molecular patterns: one is characterized by a high level of Pax6 and Emx2, accompanied by a low level of Fgf8 and Sp8, while the other displays a low level of Pax6, with a high level of Fgf8, Emx2, and Sp8 (Figure 2A; lamprey). For zebrafish, three patterns have been identified: one that exhibits a high level of Emx2 and Coup-tfi as well as a low level of Pax6 and Sp8; another where Fgf8, Emx2, and Coup-tfi are expressed highly, with Pax6 expressed at lower levels; and finally, a third pattern where Fgf8, Pax6, and Sp8 are expressed at high levels, whereas Emx2 and Coup-tfi are expressed at low levels (Figure 2A; zebrafish). For mice, there exist two molecular patterns: one is characterized by high expression of Coup-tfi and Emx2, accompanied by low expression of Fgf8, Pax6, and Sp8, while the other displays low expression of Emx2 and Coup-tfi, with high expression of Fgf8, Pax6, and Sp8 (Figure 2A; mice).

We did not observe any significant variation in the dynamic behavior of this network after the second bifurcation (*h* ≥ 2.35). On the biological side, no matter how complex the brain evolves, the binary patterning of the pallium/subpallium or olfactory bulb/olfactory cortex can always be observed in the vertebrate telencephalon [1].

### 3.4. Predictions

Based on the results above, we can make the following predictions (Figure 3A,B):

The topological structure of the core endogenous network—which determines the telencephalic anterior–posterior patterning—had already evolved in the ancestral chordate 550 million years ago.

The expression pattern of the telencephalon-like structure—in this ancestral chordate 550 million years ago and in the amphioxus today—is consistent with the stable states we obtained before the first bifurcation (0 < *h* ≤ 1.6).

An ancestral organism with a primitive telencephalic structure would develop an expression pattern consistent with the anterior part of the vertebrate telencephalon and the transition state we obtained under the parameter range of 1.65 ≤ *h* ≤ 2.325 during development, which then matured into an expression pattern consistent with the posterior part of the vertebrate telencephalon and the stable state we obtained under this parameter range of 1.65 ≤ *h* ≤ 2.325.

Experiments that reduce the cooperative effect in the network described above in vertebrates would entail a series of events: the vanishing of the anterior–posterior patterning of the telencephalon; the observation of a structure in which the expression pattern is consistent with the posterior part of the telencephalon; and the observation of a structure in which the expression pattern is consistent with the amphioxus telencephalon-like structure.

## 4. Discussion

The aim of this work is to employ this well-validated model in order to validate a first-principles-based approach to reconstructing an evolutionary trajectory. For the evolution of telencephalon anterior–posterior patterning, we found that such a five-node network alone is sufficient to draw an answer. Still, it is not enough to explain the evolution of the whole central nervous system, let alone the entire vertebrate body plan. Genes not included in this mode, such as BMP [53,54,55], Wnt [56,57], Cadherin-11 [58], Otx2 [59], and Foxg1 [60], are also important in early telencephalic development. This network can be expanded by adding new nodes and edges, which have the potential to expand into the global decision-making network for the evolution and development of the central nervous system or even the entire vertebrate body plan, as shaped by evolution.

Preliminary evidence suggests that the hypothesis that the network described in this work is conserved during evolution from the pre-vertebrate chordate to the vertebrate is plausible. The results obtained by Giacomantonio and Goodhill [26,27]—and the results described in this work—further validated this hypothesis at the computational level.

As knowledge of the real parameter dynamics is limited, if not non-existent, we used a set of dimensionless mesoscopic parameters and made the parameter sets equal within the core network level, based on a set of working hypotheses previously postulated in [21,25]. This work showed that useful predictions can still be drawn even under such little knowledge. However, exploring the range of these parameters in real biological systems is still a non-trivial task.

In response to the growing emphasis on theory–experiment coupling in the life sciences [11,12,13,14,23,24], our work suggests that the combination of existing experimental facts and the aforementioned mathematical and theoretical frameworks, although not perfect, has been able to engender new insights and verifiable predictions.

## 5. Conclusions

The current results speak to the following picture:

The ancestral chordate 550 million years ago evolved a five-node core endogenous network—described above—with weak or even negative cooperative effects, which developed a telencephalon-like structure with a similar expression pattern to the developing telencephalon during early embryonic stages in the vertebrate, but without the anterior–posterior patterning.

Between 550 and 500 million years ago, new genes emerged from genome duplication and mutations. As a result, cooperative effects within the five-node core endogenous network increased so that a primitive telencephalon structure gradually evolved to the expression pattern of the posterior part of the vertebrate telencephalon, and during development, developed an expression pattern similar to the anterior part of the vertebrate telencephalon. With the further increase in the cooperative effect during evolution, a telencephalon with anterior–posterior patterning emerged, which, like the biogenetic law suggests, would undergo a development stage with an expression pattern similar to that of the ancestral telencephalon-like structure. In short, we addressed how the telencephalon’s anterior–posterior patterning evolved quantitatively in a first-principles manner.

## Figures and Tables

**Figure 1 entropy-26-00631-f001:**
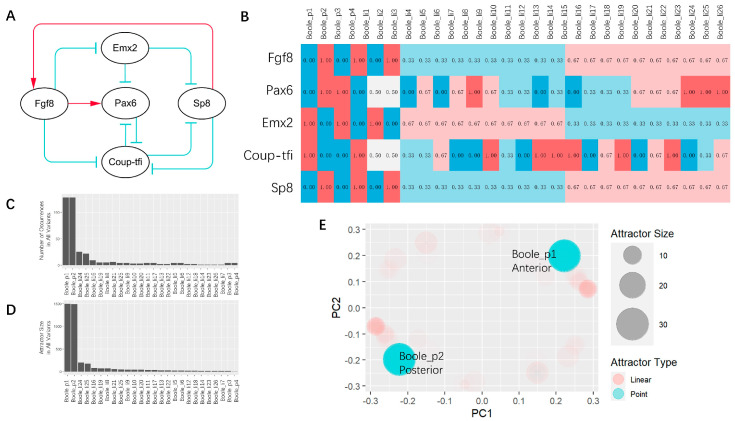
Boolean dynamics of the core endogenous network for telencephalon anterior–posterior patterning. (**A**) The network we used in this work. (**B**) Results of Boolean dynamics; limit cycles were averaged. (**C**) Occurrence of attractors among all Boolean equation variations. (**D**) Total size of an attractor in all attractors combined. (**E**) Overview of the attractors for all Boolean variations. Color of the circle represents attractor type (red: limit cycle; blue: point attractor). Aera of the circle represents attractor size. Each attractor of each Boolean variant shows the same transparency; thus, less transparency represents a higher occurrence. (_p: point attractor; _l: limit cycle.).

**Figure 3 entropy-26-00631-f003:**
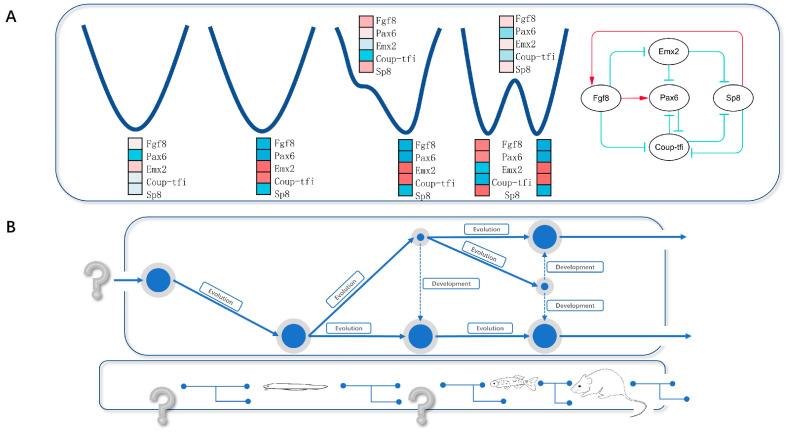
The evolutionary process of the telencephalon. (**A**) As the Hill parameter *h* increases, the number of stable states and transition states is computed by the ODE, and the expression pattern is also determined. (**B**) A schematic diagram of species involved in the evolution of the telencephalon. The representative species that evolved from chordates to vertebrates. The question marks represent our predictions: the telencephalon of the ancestral chordate was expressed earlier than that of the amphioxus; the expression pattern of the telencephalon underwent two bifurcations during the evolution of the amphioxus, corresponding to different species, including zebrafish, mice, and other possible animals.

## Data Availability

The original contributions presented in this study are included in the article. Further inquiries can be directed to the corresponding author.

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
