# Peer review of "Evolution of Telencephalon Anterior–Posterior Patterning through Core Endogenous Network Bifurcation"

_entropy, 2024, doi:10.3390/e26080631_

Round 1
Reviewer 1 Report
Comments and Suggestions for Authors
In this paper titled "Evolution of the Telencephalon Anterior-Posterior Patterning by Core Endogenous Network Bifurcation", the authors, by revisiting a gene network that explains the anterior-posterior patterning of the vertebrate telencephalon, drew an answer quantitatively of how the telencephalon anterior-posterior patterning evolved from the pre-vertebrate chordate to the vertebrate.
The historical context and references to previous research provide a solid background. The results are presented clearly, with detailed descriptions of the network dynamics and bifurcations observed. The language is precise and technical terms are used correctly. Overall, this paper is comprehensive, well-structured, and provides valuable insights into the evolution of telencephalon anterior-posterior patterning.
Here are some suggestions:
FGF8, Pax6, Sp8 expression patterns exhibit a rostral-high to caudal-low gradient, whereas Emx2, and COU-TFI (Nr2f1) expression patterns exhibit caudal-high to rostral-low gradient. Please note BMP and Wnt signaling expression patterns also exhibit caudal-high to rostral-low gradient. But this model does not use BMP/Wnt signaling.
Predictions in the paper should be supported by the presented data or references.
Author Response
|
Comments 1: FGF8, Pax6, Sp8 expression patterns exhibit a rostral-high to caudal-low gradient, whereas Emx2, and COU-TFI (Nr2f1) expression patterns exhibit caudal-high to rostral-low gradient. Please note BMP and Wnt signaling expression patterns also exhibit caudal-high to rostral-low gradient. But this model does not use BMP/Wnt signaling. |
|
Response 1: Thank you for pointing this out. We intended to develop a “first-principle” methodology to reconstruct evolutionary trajectories. On this first trial, for clarity we employed a well validated theoretical model which exhibits relatively simple and clear and dynamic behavior. We acknowledge the important role of BMP and Wnt in the early telencephalic development, which definitely should be reflected in future work.
To emphasize the point, in the revised manuscript, we have specifically addressed the issue in the Discussion section, lines 257~258, 261~264 and added relevant literature (References 53~60).
|
|
Comments 2: Predictions in the paper should be supported by the presented data or references. |
|
Response 2: Many thanks for the suggestion. We have followed the advice and searched for a series of experiments data including amphioxus, lamprey, zebrafish, and mice.
The specific revisions are presented in Figure 2 (A), cf. lines 169~174, in the Results section, lines 210~222, and added relevant literature (References 35~51).
|

Reviewer 2 Report
Comments and Suggestions for Authors
I enjoyed reading this dynamical (first principles) account of gene expression patterns during neurodevelopment (and evolution). Having said this, the text was sometimes difficult to read because of the translation to the English language. Furthermore, I think you need to orientate the reader with a summary paragraph at the end of the introduction – so that the reader understands your agenda.
I would recommend the following:
"Specifically, we address the functional specialization during neurodevelopment in terms of its underlying determinants; namely, patterns of gene expression. The cerebral cortex is segregated into functionally and architectonically distinct areas. The emergence of this segregation — during neurodevelopment — depends on the expression of several genes. Along the anterior-posterior axis, gradients of Fgf8, Emx2, Pax6, Coup-tfi, and Sp8 play a key role in functional segregation. In what follows used Boolean and dynamic models of the interactions among these five genes to reproduce the anterior-posterior expression patterns observed empirically. Furthermore, we explore different variants of interactions to see if the comparative anatomy of functional segregation (as expressed phylogenetically) can be reproduced. We first describe the two models used to account for patterns of gene expression and then consider the results in terms of the underlying dynamical attractors."
Furthermore, there are lots of grammatical errors. I have therefore converted your PDF submission into a word document. I have them gone through making suggestions using track changes. For clarity, I have accepted my deletions but have left my insertions for you to consider. You will find the above paragraph and my grammatical suggestions in the attached file (entropy-3038285-peer-reviewed.docx).
I hope that these suggestions help, should any revision be required.

Author Response
|
Comments 1: The text was sometimes difficult to read because of the translation to the English language. Furthermore, I think you need to orientate the reader with a summary paragraph at the end of the introduction – so that the reader understands your agenda. |
|
Response 1: Thank you for the suggestion. We have added a summary paragraph of the work at the end of the Introduction section, lines 58~69.
|
|
Comments 2: There are lots of grammatical errors. |
|
Response 2: Thank you for pointing this out. We greatly appreciate your help in improving the grammar. We have revised the entire English writing largely based on your editing. The full list of changes is shown below.
|
